# Molecular Aspects of Volatile Anesthetic-Induced Organ Protection and Its Potential in Kidney Transplantation

**DOI:** 10.3390/ijms22052727

**Published:** 2021-03-08

**Authors:** Gertrude J. Nieuwenhuijs-Moeke, Dirk J. Bosch, Henri G.D. Leuvenink

**Affiliations:** 1Department of Anesthesiology, University Medical Centre Groningen, University of Groningen, Hanzeplein 1, 9713 GZ Groningen, The Netherlands; d.j.bosch@umcg.nl; 2Department of Surgery, University Medical Centre Groningen, University of Groningen, Hanzeplein 1, 9713 GZ Groningen, The Netherlands; h.g.d.leuvenink@umcg.nl

**Keywords:** ischemia reperfusion injury, kidney transplantation, anesthetic conditioning, organ protection, volatile anesthetics, anesthesiology

## Abstract

Ischemia reperfusion injury (IRI) is inevitable in kidney transplantation and negatively impacts graft and patient outcome. Reperfusion takes place in the recipient and most of the injury following ischemia and reperfusion occurs during this reperfusion phase; therefore, the intra-operative period seems an attractive window of opportunity to modulate IRI and improve short- and potentially long-term graft outcome. Commonly used volatile anesthetics such as sevoflurane and isoflurane have been shown to interfere with many of the pathophysiological processes involved in the injurious cascade of IRI. Therefore, volatile anesthetic (VA) agents might be the preferred anesthetics used during the transplantation procedure. This review highlights the molecular and cellular protective points of engagement of VA shown in in vitro studies and in vivo animal experiments, and the potential translation of these results to the clinical setting of kidney transplantation.

## 1. Introduction

During the process of kidney donation and transplantation a number of potentially harmful processes will inevitably occur, which affect the viability of the graft. The donor organ is, by definition, exposed to a period of ischemia which lasts until the kidney is re-connected to the circulation of the recipient. In the case of deceased kidney donation, additional injury occurs even before graft removal. The pro-inflammatory and pro-coagulatory systemic and local renal response to brain death in the case of deceased brain dead (DBD) donors and the variable and often extended period of warm ischemia and hypoxia in the case of deceased circulatory death (DCD) donors, result in further impairments, reducing post-transplant graft function and survival. These combined effects on the graft-to-be result in a cascade of renal damage that will reveal itself at the time of transplantation when the donor kidney is reperfused in the recipient and is called ischemia reperfusion injury (IRI). Typically, IRI manifests as immediate non-function of the graft with the need for dialysis treatment until the graft recovers from the insult and starts to function. This “secondary” recovery is called delayed graft function (DGF). In cases where the injury has been too extensive to repair the transplanted kidney and it will never recover nor regain function, this is called primary non function (PNF). DGF and PNF are clinically relevant problems. Both are associated with increased morbidity, patient anxiety, prolonged hospitalization, and additional diagnostic procedures and costs. Furthermore, DGF is associated with acute rejection (AR), and the combination of DGF and AR reduces graft and patient survival [1,2].

IRI consists of a complex pathophysiology in which several molecular pathways and signaling cascades are involved [3]. It is amongst others associated with dysfunctioning of the mitochondrial respiratory chain and uncontrolled formation of reactive oxygen species (ROS) during reperfusion, leading to opening of the mitochondrial permeability transition pores (mPTP) and release of danger-associated molecular patterns (DAMPs) into the intra- and extracellular space [4,5]. From here, several injury cascades are implicated, including the activation of cell death programs such as apoptosis and (regulated) necrosis and endothelial dysfunction with loss of the glycocalyx and transmigration of leucocytes into the interstitial space. The DAMPs released upon reperfusion will further harm the graft by binding to pattern recognition receptors (PRR) such as toll-like receptors (TLR) and receptors of the complement system, leading to activation of the innate and subsequently the adaptive immune system [3]. This will increase the immunogenicity of the graft, favoring T cell and antibody mediated rejection (ABMR) and the initiation or progression of interstitial fibrosis associated with chronic graft dysfunction. In addition, protective pathways in cells are activated upon IRI by means of activation of transcription factors such as hypoxia inducible factors (HIFs) involved in the regulation of genes associated with the metabolic cell cycle, angiogenesis, and cell survival [6].

Conditioning is a broad term generally used to describe strategies inducing biochemical changes that attenuate IRI. Dependent on timing of application of the strategy, it is referred to as pre- (before ischemia), per- (during ischemia), or post- (directly upon reperfusion) conditioning. It was first described in 1986 by Murry and colleagues [7]. They reported that subjecting the heart to four brief ischemic episodes, followed by reperfusion, preceding a prolonged ischemic insult, reduced myocardial infarct size by 75% in a canine model [7]. This phenomenon is called ischemic preconditioning (IPC). Subsequently, it was found that ischemic conditioning (IC) upon reperfusion of the heart (ischemic postconditioning, IPostC) or applied to a remote tissue or organ (remote ischemic conditioning, RIC) had a similar protective effect on the myocardium [8,9]. Following the heart, (R)IC was also described for various other organs including the liver, brain, lungs, and kidney [10,11,12,13]. In addition, several non-ischemic stimuli (such as hyperthermia or transient pacing) and pharmacological substances (such as erythropoietin and nicorandil) were also found to confer cellular tolerance to a major ischemic period by underlying mechanisms similar to those mediating IC [14,15,16,17,18,19,20,21]. This is also the case for some of the generally used anesthetics agents, which is called anesthetic conditioning (AC). AC is particularly attributed to volatile anesthetic (VA) agents, such as sevoflurane, isoflurane and desflurane, and to a lesser extent to the intravenous anesthetic agent propofol. Next to this, VA affect various cells of the immune system, either by direct interaction of the drug with the cell or indirectly, by altering environmental conditions; hence IRI, of which many of these effects seem favorable during (kidney) transplantation. Therefore, the use of VA during the donation, transplantation, and even the preservation period, could be an attractive way to reduce injury and improve graft function. This review highlights the molecular and cellular protective points of engagement of VA shown in in vitro studies and in vivo animal experiments, and the potential translation of these results to the clinical setting of kidney transplantation.

## 2. Anesthetic Conditioning and Organ Protection with Volatile Anesthetic Agents

Anesthetic agents have pleiotropic effects, meaning that there are effects other than inducing anesthesia and analgesia, which affect many cells and systems in our body. This was already reported in the beginning of the 20th century. In 1911, Gaylord and Simpson showed that breast cancer cells grew more rapidly in mice when exposed to ether or chloroform [22]. A few years later, in 1916, Graham reported that leucocytes exposed to ether showed impaired phagocytosis of streptococci [23]. After these early findings, it was quiet in this area of research for a long period of time, but in the past few decades there has been a growing interest in the effects of our generally used anesthetic and analgesic agents on various organs, the immune system, and phenomena such as IRI. Experiments on pulmonary epithelial and endothelial cells suggest that the trifluoronated carbon groups, which are part of all modern VA, are responsible for the anti-inflammatory and immunomodulatory effects [24].

The first paper describing the advantageous effects of VA on IRI dates back to 1976, in which Bland and Lowenstein reported the protective properties of halothane on the myocardium in an ischemia and reperfusion (I/R) model of a canine heart [25]. In 1988, Warltier et al. showed that the recovery of cardiac function after exposing the myocardium to a brief period of ischemia was enhanced by isoflurane. Dogs anesthetized with isoflurane regained full contractile performance within two hours after reperfusion compared to control animals, which after five hours regained only 50% of their baseline contractile performance [26]. Since then, numerous animal studies and clinical trials have been published reporting the cardioprotective properties of VA against IRI and elucidating mechanistic pathways. Several meta-analyses have been performed comparing the use of VA versus propofol on cardiac and patient outcome in cardiac surgery, with results ranging from no or minor difference (e.g., lower cardiac-troponin-1,cTn1) to a reduction in cardiac complications, inotropic requirements, duration of mechanical ventilation, and mortality in the advantages of VA [27,28,29]. Modes of application of the VA in these studies ranged from specific pre- or postconditioning protocols to administration of the VA during the entire surgical procedure. The most recent and largest meta-analysis, performed by Bonnani et al. including 42 trials and 8197 patients, however, showed that VA are superior to propofol on long term mortality and postoperative morbidity [30]. In their analysis, the use of VA in patients undergoing cardiac surgery with the use of cardiac pulmonary bypass was associated with a lower one-year mortality, myocardial infarction, lower cTnT release, less need for inotropic support, shorter extubation time, and higher cardiac index/cardiac output compared to the use of propofol, indicating a myocardial protective effect of VA in this setting [30].

In addition to the heart, protective effects of VA are also described for other organs such as the lungs, liver, brain, and kidney, both in vitro and in vivo in different animal species [31,32,33,34,35,36,37,38,39,40,41]. Regarding the kidney, evidence of protection is either indirect or restricted to animal work. In rats, Lee et al. showed that clinically relevant concentrations (1 minimal alveolar concentration, MAC) of VA (sevoflurane and isoflurane) administered both during and after renal ischemia conferred profound protection against renal IRI, resulting in dramatically lower plasma creatinin levels and reduced renal necrosis 24–72 h after injury, compared with rats that received the intravenous agents pentobarbital or ketamine [42]. In mice subjected to renal I/R, anesthesia with isoflurane led to reduced infiltration of neutrophils, macrophages and lymphocytes post-reperfusion compared to mice anesthetized with pentobarbital [43]. Ko et al. studied the impact of choice of anesthetic agent on a variety of renal and hepatic function markers in patients undergoing hemi-hepatectomy [44]. Typically, during this procedure, very conservative fluid administration regimes are used. While this may achieve the aim of limiting intraoperative blood loss, it also jeopardizes renal function if significant renal hypoperfusion and ischemia occurs. In this study, patients who received desflurane had significantly better creatinine and glomerular filtration rate (GFR) values on the first day after surgery than patients anesthetized with propofol.

Mechanistically, protection against IRI with the use of VA most likely is not conferred by the activation of one single pathway. VA have been shown to interact with many of pathophysiological processes involved in IRI. These points of engagement are outlined below and summarized in Figure 1.

### 2.1. Prevention of Opening of the mPTP

Induction and opening of mPTP is an important early step in the cascade of injurious events in IRI. Opening of the mPTP will lead to the release of substances such as cytochrome C, succinate and mitochondrial DNA in the intracellular space which are able to activate cell death programs such as apotosis or (regulated) necrosis. In addition, the substances released may act as DAMPs, activating the innate immune system by binding to PRR [5]. Prevention of opening of mPTP has been shown to be an effective strategy to reduce IRI [45]. Administration of VA in the early phase of reperfusion is able to inhibit mPTP opening, most likely through multiple pathways, as displayed in Figure 2. VA are likely to reduce the activity of complex III in the mitochondrial respiratory chain, which results in an increase in superoxide (an ROS) formation. The superoxide formed reacts with nitric oxide, resulting in the formation of peroxynitrite, which itself reduces electron transport at complex I of the respiratory chain, leading to more superoxide formation [46,47,48,49]. The amounts of ROS formed in this process, however, are significantly lower than the harmful concentrations formed in the case of prolonged I/R.

In cytosol, superoxide acts as a second messenger and is able to induce translocation of protein kinase C-ε (PKC-ε) to the mitochondrial membrane [50]. Phosphorylation of PKC-ε causes opening of the K^+^-adenosine triphosphate (ATP)ase channels, resulting in an influx of K^+^ into the mitochondria, reducing the inner membrane potential. These reductions in membrane potentials have been shown to prevent mPTP production [51]. Additionally, superoxide is able to activate the reperfusion injury salvage kinase (RISK) pathway, in which the inhibition of glycogen synthase kinase 3 beta (GSK-3β) and formation of nitric oxide (NO) via endothelial nitric oxide synthase (eNOS) will lead to the prevention of mPTP opening or closing of mPTP [52,53]. The link between VA-induced activation of the RISK pathway and prevention of mPTP opening has been well established [54,55,56,57,58,59,60,61]. In contrast, the contribution of the survivor activating factor enhancement (SAFE) pathway in the VA-induced prevention of mPTP opening is less clear [62]. Superoxide may activate the SAFE pathway in which increases in anti-apoptotic B cell lymphoma 2 (Bcl- 2) activity leads to mPTP closing [63,64,65].

In primary cultured rat cardiocytes exposed to three hours of hypoxia followed by two hours of reoxygenation, Wu et al. showed that treatment with isoflurane during the hypoxic period significantly attenuated hypoxia/reoxygenation-induced apoptosis. In their experiment, isoflurane pretreatment prevented excessive ROS formation upon reoxygenation, attenuated mPTP opening, and as a result inhibited the activation of caspase 3 [66]. In a very elegant experiment, Pravdic et al. investigated the in vivo effect of isoflurane post-conditioning on cardiac mitochondria in rats exposed to cardiac I/R [67]. They showed that isoflurane protected against mPTP opening, slowed mitochondria respiration, and depolarized mitochondria. In addition, they studied the effect of isoflurane on isolated cardiomyocytes and isolated mitochondria subjected to I/R. Cells treated with isoflurane during reoxygenation showed attenuated intracellular Ca^2+^ accumulation, maintained a lower pH, and preserved the mitochondrial membrane potential during reperfusion. In mitochondria, a preserved ATP production and respiration was observed. A mildly acidotic pH at the time of reperfusion has been shown to be crucial for protection against IRI [68,69]. It has been suggested that the delayed opening of mPTP at lower pH values generates time for protective signaling pathways to be activated [68,69]. Pravdic et al. showed that the preserved acidotic mitochondrial pH after reperfusion may not be related to anaerobic metabolism during reperfusion, because isoflurane treatment also decreased mitochondrial pH under normoxemic conditions [67]. They suggest a direct effect of isoflurane on mitochondrial bioenergetics by mild inhibition of complex I, resulting in reduced H^+^ pumping out of the mitochondria or increased H^+^ influx due to mitochondrial uncoupling [70,71,72]. The study of Pravdic et al. demonstrated that post-conditioning with VA produces a significant protection of mitochondria in cardiomyocytes; whether this is also the case in renal tubular cells is unclear.

### 2.2. Protective Effects on the Glycocalyx

VA have been shown to protect the glycocalyx and to reduce endothelial injury in IRI. The glycocalyx, located on the luminal side of the vascular endothelial cells, represents a 0.1–11.0 μm layer (depending on measurement technique, localization, and species), consisting of a dynamic network of membrane-bound proteoglycans and glycoproteins [73,74,75,76,77]. The proteoglycans have a protein core, consisting of transmembrane syndecans, membrane-anchored glypicans, or secreted core proteins such as versicans, biglycans, and a various number of negatively charged glycosaminoglycan (GAG) side chains. Of these GAGs, 50–90% consist of heparan sulfate, and the remaining part of hyaluronic acid, chondroitin, dermatan and keratan sulphate [78]. Hyaluronic acid is the only GAG not covalently attached to a core protein; instead, it interacts with the endothelium by attaching to the transmembrane receptor cluster of differentiation (CD)44 [79]. It is much longer than the other GAGs and weaves through the glycocalyx [80]. Within this network of proteoglycans lie several types of glycoproteins, such as von Willebrand factor (vWF) and adhesion molecules. These adhesion molecules consist of selectins (E/Pselectin), integrins and immunoglobulins (intercellular adhesion molecule (ICAM)-1, ICAM-2, vascular cell adhesion molecule 1 (VCAM-1), platelet endothelial cell adhesion molecule 1 (PECAM-1)), which are ligands for the integrins on leucocytes and platelets, facilitating adhesion to the endothelium and transmigration to the interstitial space. Injury to the glycocalyx in the case of IRI leads to a disrupted endothelial barrier with platelet aggregation, hypercoagulabilty, inflammation, and increased vascular permeability [81]. In an isolated guinea pig heart model, hearts were subjected to 20 min of ischemia with or without 1 MAC sevoflurane administration 15 min before ischemia and/or during reperfusion. In the non-treated hearts, IRI led to a 70% increase in fluid extravasation and reduced coronary flow. In addition, increased levels of syndecan-1 and heparan sulphate were measured, indicating degradation of the glycocalyx. Sevoflurane treatment, pre- and post-conditioning, attenuated these changes and protected the glycocalyx from shedding [82]. The authors attribute this effect to lysosomal membrane stabilization by sevoflurane, with a consequent lower release of cathepsin B. This lysosomal protease is associated with degradation of the extracellular matrix and is released upon injury. Lower cathepsin B levels were measured in the treated groups [82]. The same research group showed that 1 MAC sevoflurane preconditioning led to reduced (to near baseline levels) cellular adhesion of infused human leukocytes and platelets in an isolated guinea pig heart model subjected to 20 min of warm ischemia and 10 min of reperfusion. In the non-treated group, 39% of the infused leukocytes and 25% of the platelets adhered to the endothelium, respectively. In the treated group, this was 22% and 12%, respectively. Again, the non-treated group showed increased levels of syndecan-1 and heparan sulfate. Electron microscopy showed a preserved integrity of the glycocalyx in the treated group [83]. In addition, in an isolated guinea pig model with hearts perfused with a modified Krebs-Henseleit buffer, perfused with a oxygenated pulsatile flow at 70 cm H_2_O and subjected to 20 min of warm ischemia followed by 40 min of reperfusion, Chen et al. showed that an addition of sevoflurane at 2% (1 MAC) to the gas mixture before ischemia and during reperfusion strongly inhibited the shedding of hyaluronic acid during the entire period, even prior to ischemia [84]. Treatment with sevoflurane completely prevented the increase in coronary leak seen in the non-treated group. Electron microscopy revealed a distinct glycocalyx present in the treated group, whereas in the non-treated group the glycocalyx could not be visualized. Global oxidative stress measured with the purine/uric acid ratio did not differ between groups. Next to this, there was no difference between groups in lactate washout, indicating that the ischemic impact on the hearts was similar in both groups and sevoflurane treatment seems not to influence this. Local effects at the vessel wall, however, cannot be excluded. In an in vivo pig model subjected to 90 min of ischemia induced by balloon insufflation in the thoracic aorta, followed by 120 min of reperfusion, Annecke et al. compared the effect of sevoflurane anesthesia to a propofol anesthesia at clinically relevant concentrations on unmeasured anions (represented by the strong ion gap (SIG), predictive of negative outcome) and shedding of heparan sulphate. In the propofol-treated animals, SIG significantly increased after reperfusion, which was not the case in sevoflurane-treated animals. In both groups, levels of heparan sulphate increased after reperfusion. In the sevoflurane animals, this increase stabilized after 120 min of reperfusion, whereas in the propofol-treated animals shedding of HS continued to rise, suggesting a superior role of sevoflurane over propofol in protection of the endothelial glycocalyx [85]. Modern organ preservation of kidney grafts has shifted from cold storage to hypothermic machine perfusion and has now entered an era of normothermic oxygenated machine perfusion. Based on the obtained knowledge described above, the addition of a VA to the perfusate might be a potential therapeutic strategy to protect the integrity of the glycocalyx and reduce exposure of endothelial-bound proteins such as vWF and adhesion molecules.

### 2.3. Upregulation of Hypoxia Inducible Factors (HIFs)

HIFs are heterodimeric transcription factors protecting the cell against IRI by the regulation of genes involved in metabolic cell cycle, facilitating cellular adaptation to low oxygen conditions. Under normal conditions, the amino acid proline on the α subunit is rapidly hydroxylated by oxygen-sensing prolylhydroxylases (PHD), inducing conformational changes enabling von Hippel-Lindau tumor suppressor protein (pVHL) to bind with the α-subunit, leading to degradation of the complex. Ischemia/hypoxia inhibits the PHD, which enables nuclear translocation of the α-subunit, binding of the α-subunit to the β-subunit, and the formation of HIF. In the nucleus, HIF will bind with the hypoxia response promotor element (HRE) and the transcription of various protective genes, such as adenosine receptors, VGEF and erythropoietin. There are two types of α-subunits, HIF-1α and HIF-2α, which have common but also specific target genes. In the kidney, HIF-1α is predominantly located in glomerular and tubular cells, whereas HIF-2α can be found in glomerular cells, endothelial cells, and fibroblasts [86,87,88]. Several studies suggest a protective role of HIFs during renal IRI. Oda et al. showed that increased expression of HIF-1α after reperfusion was associated with improved graft outcome in deceased donor kidney transplant recipients [89]. Kojima et al. showed, in a renal I/R mouse model, that HIF-2α^−/−^ mice showed increased renal injury compared to HIF-2α^+/+^ mice. Along with the direct protective effects of HIFs by transcription of protective genes, increasing evidence suggests the role of HIF activation in the modulation of other protective signaling pathways [90].

The ability of VA to upregulate HIF-1α and HIF-2α is being considered as one of the underlying mechanisms of AC. In an isolated I/R rat heart model, Yang and colleagues showed that 15 min of sevoflurane exposure upon reperfusion resulted in an upregulation of HIF-1α. Sevoflurane-treated hearts showed improved mitochondrial function and structure and improved cardiac function compared to non-treated hearts [91]. These protective effects were abolished if the hearts were treated with 2-methoxyestradiol, an HIF-1α inhibitor [91]. In a renal I/R model, wild type mice pretreated with sevoflurane showed significantly lower blood urea nitrogen (BUN) and creatinine levels and higher HIF-2α expression compared to wild type mice without sevoflurane pretreatment and HIF-2α^(−/−)^ knock out mice pretreated with sevoflurane [92].

Zhao and colleagues report interesting findings of the influence of xenon (Xe), a potent anesthetic gas, on HIF-1α upregulation in an in vitro model of human proximal tubular cells (PTC) and an in vivo rat transplant model [93]. Human PTCs were subjected to a 24 h period of hypoxic cold preservation and pre- (24 h before hypoxia) or post-conditioning (upon reperfusion) by a 2 h exposure of a gas mixture containing 70%Xe–5%CO_2_–25%O_2_. Both groups showed upregulation of HIF-1α, VGEF and Bcl-2 compared to non-treated PTCs. Furthermore, the Xe-treated cells showed an improved cytoskeleton, significantly lower translocation of high mobility group box-1 (HMGB-1) from the nucleus into the cytosol, and less activation of TLR4. These results were abolished when the gene silencer HIF-1α siRNA was administered. In their iso- and allograft transplant model, rats were exposed to a 70%Xe–30%O_2_ or 70%N_2_O–30%O_2_ mixture 24 h before organ retrieval (donor) or directly upon reperfusion (recipient). Kidneys underwent cold storage at 4 °C for 16(allografts)–24(isografts) h. Both pre- and post-conditioning resulted in reduced nuclear translocation of HMGB-1 and TLR4 activity as well as lower serum levels of HMGB-1. Xe treatment led to decreased NF-κB and caspase 3 expression, and serum levels Il-1β, tumor necrosis factor α (TNF-α), IL-6 were reduced. Again, the addition of siRNA HIF-1α abolished the renoprotective effects. Graft survival was significantly prolonged in the Xe-treated groups in both transplant models, and Xe-treated allografts showed reduced CD3+ T cell infiltration at day 20. Taken together, the HIF-1α upregulation could have induced nuclear and cytoskeletal stability leading to decreased HMGB-1/TLR4 signaling, resulting in lower expression of NF-κB and caspase activation. The authors suggested the Xe induced phosphoinositol 3 kinase (PI3K)- protein kinase B (Akt)- mammalian target of rapamycin (mTOR) pathway to be responsible for the upregulation of HIF-1α but could not confirm this [93]. In several studies in myocardial or cerebral cells, however, the upregulations of HIF-1α by sevoflurane and isoflurane via the PI3K-Akt-mTOR pathway have been shown [94,95]. Propofol, on the other hand, has been shown to abolish HIF-1α upregulation by isoflurane or inhibit HIF-1α production itself [79,96,97].

### 2.4. Effect on Renal Tubular Cells and Sphingosine-1-Phosphate Signaling Pathway

Lee et al. performed extensive research on the impact of VA on I/R injured renal tubular cells in several in vitro and animal experiments (Figure 3). They showed that VA exposure induces translocation of phosphatidylserine (PS) to the outer leaflet of the plasma membrane [98]. This externalization of PS inflicts a release of transforming growth factor-β (TGF-β) in neighboring cells via the ligation of PS receptors. Externalization of PS is a strong apoptotic signal for macrophages to engulf these cells. In addition, PS externalization entails an increase in caveolae formation in the cell membrane with sequestration of key signaling proteins such as TGF-β receptors, extracellular regulated kinase (ERK), sphingosine kinase 1 (SK-1), and sphingosine-1-phosphate [40,99,100,101]. Binding of TGF-β to the TGF-β receptor results in increased expression of CD-73 via the nuclear translocation of transcription factor mothers against decapentaplegic homolog 3 (SMAD-3). This increased CD-73 expression enhances the formation of adenosine, a well-known potent anti-inflammatory mediator. Adenosine induces protection upon reperfusion by activating, amongst others, the RISK and the SAFE pathways [102,103,104]. Lee et al. showed that, in renal tubular cells, activation of adenosine receptor (A1AR) by adenosine results in sphingosine kinase (SK-1) upregulation. This occurs directly via hypoxic inducible factor 1α (HIF-1α) signaling, or indirectly via increased IL-11 synthesis through the activation of the ERK/mitogen-activated protein kinase (ERK/MAPK) signaling pathway [105,106,107]. SK-1 itself promotes sphinogosine-1-phosphate (S1P) synthesis, which targets the S1P-receptor (S1PR), a G protein-coupled receptor which is sequestered in the caveolae. S1P-S1PR signaling is associated with cell survival and cell growth via the RISK and SAFE pathways [105,108,109,110,111,112,113]. Lee et al. showed that, in renal cells, S1P–S1PR signaling plays an important role in VA-induced protection. Blockade of the S1PR with a selective antagonist abolished the protective effect of isoflurane in renal tubular and endothelial cells, as was the case for SK-1 inhibitors [114]. The role of S1P signaling in renal IRI and transplantation is increasingly recognized. In mice subjected to renal I/R, treatment with the selective S1PR agonist FTY720 attenuated IRI [115]. In a renal transplant model, rats treated with FTY720 reduced apoptosis and increased the proliferation of renal tubular cells [116].

### 2.5. Effect on Circulating Immune Cells

We have described the role of specific cells and systems of the innate and adaptive immune system in IRI and kidney transplantation [3]. IRI is accompanied by sterile inflammation in which the innate as well as the adaptive immune systems are involved. DAMPs are endogenous molecules released from injured or dying cells or the extracellular matrix. They can originate from the nucleus, cytosol, cell membrane, mitochondria, etc., and can be protein or non-protein. DAMPs are an important trigger for activation of the innate immune system by binding to PRRs such as TLRs and receptors of the complement system. This already starts in the donor [117,118]. According to the “danger model”, the presence of danger, reflected in cell stress or injury and thus release of DAMPs, is the major factor that induces (allo)immunity, rather than the presence of non-self-tissue [119,120]. The impact of IRI on the activation of both T and B cells in driving humoral injury and rejection in organ transplantation is therefore increasingly recognized [121,122].

VA affect many cells of the immune system. This is either by direct interaction of the drug with the cell, or indirectly by altering environmental conditions; hence, IRI. Many of these effects, summarized in Table 1, seem favorable during (kidney) transplantation. Most studies addressing the effects of anesthetic agents on the immune system, however, are in vitro experiments or animal studies. Unfortunately, these results are not directly translational to the clinical setting; however, clinical trials looking into the effect of different anesthetic agents or techniques on circulating immune cells, mostly in the field of oncological surgery, are emerging (NCT03193710, NCT03431532, NCT02567942).

#### 2.5.1. Innate Immunity

The innate, or non-specific, immune system is evolutionary the oldest part of the immune system. It acts on injury with a fast, short-lasting, and non-specific response, and comprises different cells and systems. Members of this response, such as neutrophils and macrophages, are already involved in the early stage of IRI.

Neutrophils are regarded as primary mediators of injury. Upon reperfusion, these cells adhere to the endothelium and migrate into the graft where they release proinflammatory cytokines, such as interleukin (IL)-4, IL-6, TNF-α, interferon (IFN)-γ, ROS and proteases, leading to injury of the graft and chemotaxis of other immune cells [123]. VA have shown to impair the function of neutrophils and decrease the number of reacting cells. In an in vitro experiment, human neutrophils were exposed to several VA at various concentrations (0.5, 1.0 and 2.0 MAC) and were stimulated with bacterial peptides N-formyl-L-methionyl-L-leucyl-phenylalanine (nFMLP) and phorbol-12-myristate-13-acetate (PMA). The presence of halothane, enflurane, and sevoflurane induced an increase in the activation threshold on FMLP stimulation, correlating with reduced H_2_O_2_ (ROS) production. Isoflurane had no effect. The presence of desflurane, however, increased H_2_O_2_ production of neutrophils two-fold, followed by transient suppression of neutrophil function [124]. At clinically relevant concentrations, sevoflurane and isoflurane have been shown to decrease neutrophil adhesion to endothelial cells by inhibiting activation of these neutrophils [125]. In a liver transplantation model in rats, sevoflurane anesthesia attenuated neutrophil-mediated renal injury and was associated with decreased neutrophil infiltration and lower levels of TNF-α and IL-6 compared to chloral hydrate anesthesia [126].

Another subset of innate immune cells involved in the early stage of IRI are monocytes and their differentiates, macrophages. Monocyte–macrophages can develop in two distinct subsets, depending on the stimulus. Classically activated macrophages (M1) are induced by cytokines such as IFN-γ and microbial products, and are involved in the inflammation and phagocytosis of bacteria, etc. Alternatively, activated macrophages (M2) are induced by IL-4 and Il-13 produced by T helper 2 (Th2) cells and other leucocytes. M2 cells function to control inflammation and are involved in tissue repair and fibrosis [127]. Upon activation, M1 cells release proteolytic enzymes and proinflammatory cytokines such as IL-1β, IL-6, IL-8 and TNF-α, and IFN-γ [128]. The majority of VA have suppressing effects on monocytes and macrophages. Sevoflurane and isoflurane are associated with reduced cytokine concentrations of TNF-α, IL1, IL6, IL8 and IL10 [129,130]. The release of IFN-γ, a T helper 1 (Th1) cytokine, is involved in the expression of inducible nitric oxide synthase (iNOS) [131,132]. Numerous in vivo and in vitro investigations have demonstrated that the inhibition or absence of iNOS was associated with a reduction in IRI [133,134]. The effect of VA on iNOS and subsequently on NO production is dependent on different circumstances and the type of VA used. Sevoflurane attenuated NF-κB activation and reduced the expression of iNOS, which was accompanied by a decrease in infarct size and creatine kinase release [135]. In an in vitro experiment, Zha et al. compared the effects of isoflurane, sevoflurane and propofol on macrophage phagocytosis. They demonstrated that isoflurane (1%), as well as sevoflurane (1.5%), reduced phagocytosis by 50%, while propofol had no significant effect on macrophage phagocytosis [136]. Photo labeling showed that sevoflurane bound to Ras-related protein 1 (Rap1), affecting Rap1 activation, which appeared to be crucial in initiating phagocytosis [136].

Natural killer (NK) cells are the only cells of the innate immune system that originate from the lymphoid cell lineage. Their role in oncological surgery is extensively studied, because NK cells play an important role in first line defense against tumor cells [137]. Their role becomes clearer in kidney transplantation and renal IRI. Zhang and colleagues showed that NK cells can induce tubular cell death in vitro, possibly by interaction of retinoic acid early inducible 1 (RAE-1) on renal tubular cells and natural killer group 2D (NKG2D) receptors on NK cells [138]. In a mouse model, they showed that NK cells quickly infiltrate into the injured kidney following I/R, and that NK cell depletion was protective in this renal IRI model [138]. NK cells have been proven to be directly responsible for tubular epithelial cell injury, which makes them a major contributor to renal IRI [139]. A specific subset of NK cells, invariant NKT (iNKT) cells, infiltrate the kidney 30 min after reperfusion and lead to a significant release of pro-inflammatory cytokines of Th1 cell type (IFN-γ, TNF-α) and Th2 cell type (IL4, IL13) [140]. Blocking of iNKT has shown to prevent acute kidney injury (AKI) after IRI [137]. VA have been shown to suppress NK cell cytotoxicity and cytokine-associated NK cell activation [141,142,143]. Several in vitro studies have demonstrated a decreased response of isoflurane-treated NK cells to IFN-γ, while sevoflurane is associated with a decreased release of TNF-α [144,145,146]. Propofol, on the other hand, seems to preserve or enhance the cytotoxicity of NK cells, as shown in an ex vivo stimulation study of NK cells derived from patients with gastric cancer and anesthetized with either propofol or sevoflurane. Propofol, in contrast to sevoflurane, was shown to enhance the cytotoxicity of NK cells [147]. A meta-analysis of the impact of anesthetic exposure on NK cell function, however, showed significant data heterogenicity and was without a conclusive association.

#### 2.5.2. Adaptive Immunity

In the case of IRI, CD4+ Th cells as well as CD8+ cytotoxic T cells are found in the kidney and are important mediators of IRI [148,149,150,151]. T cell-deficient mice showed attenuated renal IRI, and adoptive T cell transfer experiments in athymic mice resulted in acute kidney injury (AKI) [152,153,154]. T cell activation occurs through binding of the T cell receptor (TCR) with a major histocompatibility complex (MHC) on the antigen presenting cell (APC). After activation, T cells proliferate and differentiate, and might harm the graft through cytokine-mediated inflammation. The effector CD4+ T cells can differentiate into three subtypes; type 1 (Th1), type 2 (Th2) and type 17 (Th17). Of the three CD4+ subtypes, Th1 and Th17 responses are most deleterious. Blockade of C–C chemokine receptor type 5 (CCR5), a receptor for chemokine (C–C motif) ligand 5 (CCL5), which is released by Th1 cells, resulted in the protection of kidney IRI in mice models, whereas in a study of Guo et al., Th17 response was increased and activated by an NF-κB pathway which resulted in aggravated kidney injury after IRI [155,156]. The Th2 response entails anti-inflammatory cytokines, which are known to have protective properties against IRI. In mice models without IL4, a mediator of Th2 response, kidney injury was significantly more pronounced [157]. In general, exposure to VA leads to a decreased number and proliferation of T cells. However, not all VA have a comparable effect on T cell response. Conflicting data about the effect of VA on the Th1/Th2 ratio have been published over the years. Exposure to isoflurane did not result in a changed Th1/Th2 ratio in a study of Ren at al., but it was decreased in a study by Inada et al. [158,159]. Sevoflurane was associated with a decreased Th1 concentration, and consequently, a decreased Th1/Th2 ratio [143]. Finally, no differences in Th1/Th2 ratio could be demonstrated with the use of desflurane [160].

Regulatory T cells (Tregs) suppress excessive immune responses and have a potentially promising role in the reduction in IRI [161,162]. Cao Jun et al. concluded that expanded Tregs participated in the repair of the early phase of renal IRI [163]. Little is known about the effects of VA on Tregs. In patients having breast cancer surgery, equipotent doses of propofol and sevoflurane did not result in different CD39, CD73 Treg populations 1 or 24 h post-surgery [164]. In a randomized controlled trial in LDKT, comparing sevoflurane to desflurane, the use of desflurane induced an increase in peripheral blood Tregs 24 h post-transplantation [165].

VA might have apoptotic effects on lymphocytes; although consequently they no longer are able perform their immunological function. Isoflurane and sevoflurane directly induce apoptosis in a dose-dependent manner in human peripheral lymphocytes [166]. The induction of apoptosis is accompanied by the increased caspase 3 activity in lymphocytes [167].

While the role of the T cell may either be protective or pathogenic depending on the subtype of T cell, the role of the B cell in all circumstances is pathogenic during IRI. In a study of Jang et al., B cell trafficking was observed into the allograft during warm kidney IRI and altered tubular repair process [168]. Moreover, the infiltration of B cells is involved in allograft rejection and poor graft survival [169]. Sevoflurane is associated with a decreased number of peripheral blood lymphocytes and splenic B cell counts [170]. However, no differences in the number of B cells were observed between patients receiving propofol or sevoflurane for laparoscopic hysterectomy [171].

#### 2.5.3. Effect on Lymphocyte Function Antigen-1

Yuki et al. showed that clinically relevant concentrations of isoflurane and sevoflurane are able to inhibit Lymphocyte Function Antigen-1 (LFA-1) in vitro [172,173]. LFA-1 belongs to the integrin family, a family of adhesion molecules consisting of α and β subunits. LFA-1 (α1β2) is expressed in all leukocytes. Upon activation of the leukocyte by chemokines or antigens, LFA-1 undergoes conformational changes, facilitating its ability to bind with its ligand, via a process which is called inside-out signaling. The most important ligand for LFA-1 is ICAM-1, of which expression on endothelial cells, APCs and other cells is increased upon IRI. Interaction of LFA-1 with endothelial ICAM-1 assures firm adhesion of the leukocyte to the endothelial cells, and therefore facilitates transmigration of the leukocyte into the interstitium [174]. In the case of NK cells, LFA-1–target cell ICAM-1 interaction is necessary for activation of the NK cell and lysis of the target cell [175,176]. LFA-1 activation on T cells facilitates APC binding and is involved in T cell activation (Figure 4). VA agents inhibit LFA-1 at the lovastatin binding site at the α subunit, keeping the molecule in the inactive state, and therefore preventing binding of LFA-1 with its ligand [172,173]. Short term blockade of LFA-1 shows promising results in pancreatic islet transplantation in different animal models [177,178,179]. In humans, efalizumab (Raptiva^®^, Genentech, Merck Serono), a human monocolonal IgG LFA-1 antibody, has been studied in renal transplantation in order to reduce rejection. Efalizumab decreased acute rejection but increased the incidence of post-transplant lymphoproliferative disorder (PTLD) [180]. Due to the presentation of four cases of progressive multifocal leukoencephalopathy in psoriasis patients treated with efalizumab, the drug was taken off the market in 2009 [181]. LFA-1 still remains an attractive target in (kidney) transplantation, not only in the light of immune therapy but also in the reduction in IRI. The question remains as to whether a short-term exposure to VA, only during the transplant procedure, is able to exert beneficial effects on the kidney graft via LFA-1 inhibition. Propofol is also able to inhibit LFA-1 in vitro, but it is unknown whether this is the case in vivo [182]. Extrapolation of in vitro experiments with propofol to the clinical setting might not be straightforward because protein binding of propofol in whole blood is significant, and free propofol concentrations might be significantly lower compared to the concentrations used in in vitro experiments [183].

## 3. Clinical Trials on the Potential of Volatile Anesthetic Protection in Kidney Transplantation

Only a few clinical trials have looked into the effect of anesthetics used in organ procurement procedures and kidney transplantation on graft outcomes. In a retrospective cohort analysis, Perez-Protto et al. studied the impact of VA during DBD donor organ procurement on graft survival in recipients (preconditioning) and compared VA-exposed (*n* = 138) to non-VA-exposed (*n* = 75, no anesthetic, etc.) donors [184]. They found no significant differences between the groups in 30-day and five-year graft survival of heart, liver, lung, and kidney transplants. In this analysis, however, sample size was relatively small, and the rates of graft failures in both groups were low (25 of the 446 transplanted organs, 5.6%); consequently, there was a lack of power to confirm conclusions. In addition, the dosage and duration of exposure of the VA were unclear. In prospective cohort analysis, Lee et al. compared kidney function in recipients of living donor kidneys according to the type of anesthetic used in their matching donor (preconditioning) and found no differences between desflurane (*n* = 50) and propofol (*n* = 49) [185]. Changes in serum creatinine post-transplantation and estimated glomerular filtration rate (eGFR) at day of discharge were comparable between groups. In a pilot proof of concept study of the Volatile Anesthetic Protection of Renal transplants (VAPOR) project of our group, we compared a sevoflurane-based anesthesia with a propofol-based anesthesia in living donor kidney transplantation (LDKT) (VAPOR-1 study; pre- and post-conditioning) [186]. Although this specific setting of kidney transplantation is associated with significantly less IRI, low incidence of DGF (<5%) and better graft outcome compared to deceased donor kidney transplantation, it provides an elegant research model due to a homogeneous model of IRI and reproducible ischemia times. In addition, this model provides the possibility to treat the donor. The primary outcome measure was a set of urinary biomarkers consisting of kidney injury molecule-1 (KIM-1), N-acetyl-D-glucosaminidase (NAG), and heart-type fatty acid binding protein (H-FABP) reflecting kidney injury with the hypothesis of higher levels in propofol-treated patients. Whereas H-FABP was highest in the first urine sample upon reperfusion and levels were comparable between treatment groups, levels of KIM-1 and NAG increased again the first day after transplantation after an initial decrease. To our surprise, sevoflurane-treated patients showed higher levels of KIM-1 (second day) and NAG (first day, second day) after transplantation compared to propofol-treated patients. This, however, was not associated with inferior graft outcome [182]. In contrast, higher KIM-1 levels the first day after transplantation were significantly associated with better eGFR one month post-transplantation (*R* = 0.412, *p* = 0.002), which was almost the case for KIM-1 levels measured the second day after transplantation (*p* = 0.074). This could be due to the opposed dual role of KIM-1, with evidence that in acute renal injury, KIM-1 plays a role in the regeneration and repair process. AKI surviving PTC expressing KIM-1 are able to phagocytize luminal cellular debris consisting of apoptotic and necrotic cells, enabling the PTC to down-regulate the innate immune response upon AKI, which could be beneficial in kidney transplantation [187,188]. Higher NAG levels (first and second in this case) might be a reflection of regenerated tubular cells showing baseline lysosomal activity rather than a reflection of injury. An intriguing outcome of our study was the difference in incidence of T cell mediated rejection during a two-year follow-up in favor of sevoflurane-treated patients. This, however, was not our primary outcome measure, and the number of events and patients (*n* = 9, *n* = 57, respectively) did not allow us to perform a multivariate analysis. Well-designed randomized controlled trials (RCTs) need to be performed to see whether the effects of VA are clinically relevant and are attributable to improved outcome after kidney transplantation. One can hypothesize that the potential beneficial effects are more pronounced in the setting of deceased donor kidney transplantation, because the extent of IRI in this setting exceeds the amount encountered in LDKT. Currently, VAPOR-2, an international multicenter RCT (ClinicalTrials.gov: NCT02727296), in which 488 patients receiving a DBD or DCD kidney will be randomized to a sevoflurane–remifentanil-based anesthesia or a propofol–remifentanil-based anesthesia (post-conditioning), is running. The primary outcome in this study is DGF, an adequate reflection of IRI. After completion of the VAPOR project, we hope to be able to answer the question as to whether the choice of anesthetics contributes to improved graft and patient outcome after kidney transplantation.

Which type of VA confers the most protective effect is disputed. In the animal experiments of Lee et al., isoflurane and sevoflurane showed comparable effects but were significantly more profound than desflurane [42]. This could be due to the fact that desflurane has a lower lipid solubility. Interaction of the VA with the bi-lipid membranes of cells and organelles has been proposed as a potential point of engagement of the protective effects of these agents. Park et al. retrospectively compared the effect of desflurane (*n* = 71) and sevoflurane (*n* = 73) in living kidney donors and their matching recipients (pre- and post-conditioning). They found no differences between both anesthetics in terms of eGFR, creatinine and blood urea nitrogen (BUN) one year post-transplantation, as was the case for DGF, AR and graft failure [189]. Similar findings were found by Savran Karadeniz et al., in a prospective cohort analysis comparing desflurane (*n* = 30) and sevoflurane (*n* = 35) [190]. Although sevoflurane anesthesia was associated with higher IL-8 levels one day and one week after transplantation, both agents had similar effects on graft function [190].

## 4. Summary

Animal and in vitro experiments in various organs demonstrate that VA interfere with many of the processes underlying the pathophysiology of IRI encountered in kidney transplantation. Introduction of VA before ischemia or directly upon reperfusion can prevent opening of the mPTP, an important initiating step in the injurious cascade of IRI. In addition, VA have been shown to prevent degradation of glycocalyx and preserve this important barrier of the endothelium. Both pre- and post-conditioning were found to be associated with upregulation of the HIFα subunit and improved organ function after I/R. Above this, VA affect many cells of the innate as well as the adaptive immune system, either by direct interaction with the cell or indirectly by altering environmental conditions; hence, IRI. Many of these effects seem favorable during (kidney) transplantation. Altogether, VA potentially could have a protective effect against the consequences of IRI and improve graft outcome in kidney transplantation. These effects, however, are dose-, time-, organ-, and context-dependent, and are not necessarily translational to clinical practice. It is key for future benefit of our patients that we exceed with well-designed clinical trials to explore whether these protective effects are clinically relevant in patients with all kinds of comorbidities, multiple medications, and genetic variation, while at the same time proceeding with further unraveling of underlying mechanistic pathways.

## Figures and Tables

**Figure 1 ijms-22-02727-f001:**
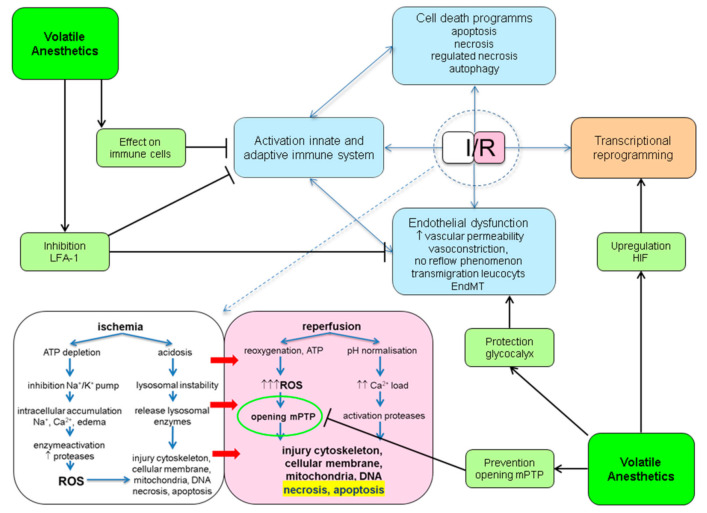
Potential protective points of engagement of volatile anesthetics (VA) in ischemia reperfusion injury (IRI). ATP: adenosine triphosphate; EndMT: endothelial to mesenchymal transition; HIF: hypoxic inducible factor; I/R: ischemia/reperfusion; LFA-1: lymphocyte function antigen-1; mPTP: mitochondrial permeability transition pore; ROS: reactive oxygen species.

**Figure 2 ijms-22-02727-f002:**
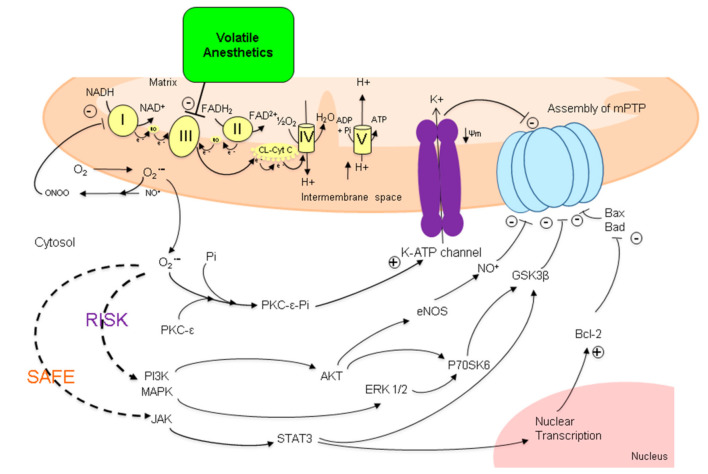
Proposed effects of volatile anesthetics on the level of mitochondria. VA are likely to reduce complex III activity in the mitochondrial respiratory chain, resulting in an increase in superoxide (an ROS) formation. Superoxide then may react with nitric oxide, resulting in the formation of peroxynitrite, which itself reduces electron transport at complex I leading to more superoxide formation. From here, several pathways can be induced, ultimately resulting in reduced formation of mPTP (via PKC) or the prevention of opening of mPTP (via RISK or SAFE pathways). NAD: nicotinamide–adenine–dinucleotide; FAD: flavin–adenine–dinucleotide; CL-Cyt C: cytochrome C oxidase; PKC-ε: protein kinase C-ε; Pi: phosphate; Ψm: membrane potential; mPTP: mitochondrial permeability transition pore; PI3K: phosphoinositol 3 kinase; MAPK: mitogen activated kinse; JAK: Janus kinase; AKT: protein kinase B; eNOS: endothelial nitric oxide synthase; NO: nitric oxide; ERK1/2: extracellular signal-regulated kinases1/2; P70SK6: ribosomal protein S6 kinase beta-1; GSK3β: glycogen synthase kinase 3 beta; Bcl-2: B cell lymphoma 2; Bax: bcl-2-like protein 4; Bad: BCL2 associated agonist of cell death; RISK pathway: reperfusion injury salvage kinase; SAFE pathway: survivor activating factor enhancement pathway.

**Figure 3 ijms-22-02727-f003:**
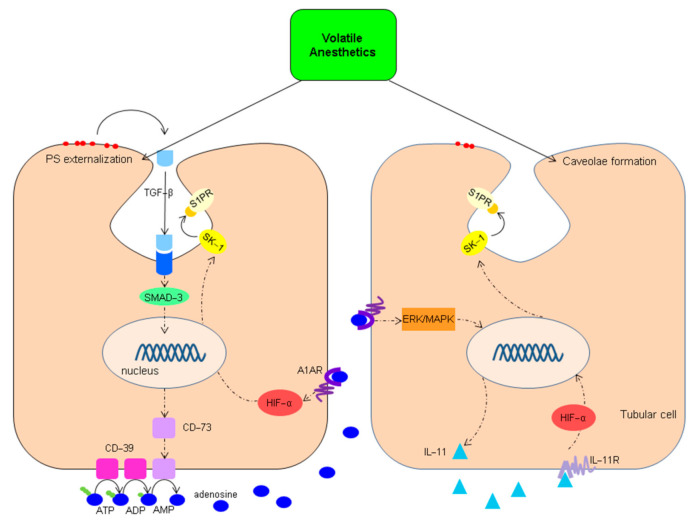
Proposed effect of volatile anesthetics on renal tubular cells and the sphingosine-1-phosphate signaling pathway. Exposure of renal tubular cells leads to the externalization of phosphatidylserine (PS) with a release of transforming growth factor-β (TGF-β) and the formation of caveolae with sequestration of key signaling proteins such as TGF-β receptors, extracellular regulated kinase (ERK), sphingosine kinase 1 (SK-1) and sphingosine-1-phosphate (S1P). SMAD-3: mothers against decapentaplegic homolog 3; CD: cluster of differentiation; HIF-α: hypoxic inducible factor-α; A1AR: A1 adenosine receptor; IL-11: interleukin 11; IL-11R: IL-11 receptor. Adapted from: Fukazawa K, Lee HT. Volatile anesthetics and AKI: risks, mechanisms, and a potential therapeutic window. J. Am. Soc. Nephrol. 2014 May; 25(5):884–892. doi:10.1681/ASN.2013111215.

**Figure 4 ijms-22-02727-f004:**
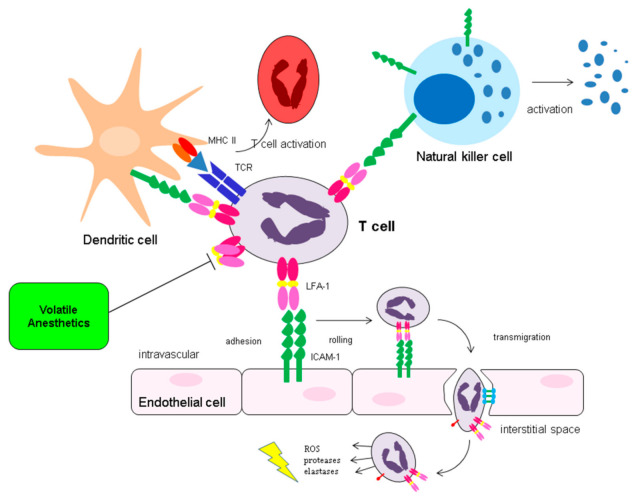
Interaction of LFA-1 with its ligands and consequences. Upon activation of the leukocyte by chemokines or antigens, lymphocyte function antigen-1 (LFA-1) undergoes conformational changes, facilitating its ability to bind with its ligand, via a process which is called inside-out signaling. The most important ligand for LFA-1 is intercellular adhesion molecule 1 (ICAM-1), of which expression on endothelial cells, antigen presenting cells (APCs) and other cells is increased upon IRI. Interaction of LFA-1 with endothelial ICAM-1 assures firm adhesion of the leukocyte to the endothelial cells and facilitates transmigration of the leukocyte into the interstitium. In the case of natural killer (NK) cells, LFA-1–target cell ICAM-1 interaction is necessary for activation of the NK cell and lysis of the target cell. LFA-1 activation on T cells facilitates APC binding and is involved in T cell activation. VA are able to inhibit activation of LFA-1, keeping it in its folded state. MHC-II: major histocompatibility complex II; TCR: T cell receptor.

**Table 1 ijms-22-02727-t001:** Effects of volatile anesthetics on various cells of the immune system.

Cell Type	Effect
Innate immune system	
Neutrophils	↓ cellular function ↓ ROS production ↓ expression of endothelial adhesion molecules ↓ adhesion to endothelium ↓ tissue infiltration
Monocytes/macrophages	↓ number ↓ release proinflammatory cytokines IL-1β, TNF-α, IL-6, IL-8 ↑ expression iNOS and NO production Influence on APC function unknown
Natural Killer cells	↓ cytotoxicity ↓ release proinflammatory cytokines
Dendritic cells	Unknown
Adaptive immune system	
T cells	↓ number and proliferation ↓ Th1/Th2 ratio Induction apoptosis ↓ release proinflammatory cytokines ↓ adhesion molecules
B cells	↓ number Induction B cell injury
T regs	Unknown

ROS: reactive oxygen species; IL-1β: interleukin 1β, TNF-α; tumor necrosis factor-α; IL-6: interleukin 6; IL-8 interleukin 8; iNOS: inducible nitric oxide synthase; NO: nitric oxide; APC; antigen presenting cell; Th1: T helper cell 1, Th2: T helper cell 2; T regs: regulatory T cell.

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
