# Peer review of "Molecular Aspects of Volatile Anesthetic-Induced Organ Protection and Its Potential in Kidney Transplantation"

_ijms, 2021, doi:10.3390/ijms22052727_

Round 1

Reviewer 1 Report

This is a review describing the effects of volatile anesthetic agents on cells.

The authors discuss the effects of VA on cells extensively, although their intention seems to be to focus on the effects on grafts in kidney transplantation.

In this paper, the authors discuss the effects of VA on cells in a wide range of conditions, including pre- (before ischemia), per- (during ischemia) or post- (directly upon reperfusion) conditioning.

This is a comprehensive review that includes the latest papers on VA conditioning. But I think there are several issues to be approached by the authors.

The followings are my comments on the manuscript.

#
Explicitly distinguishing between "pre-", "per-", and "post-" in the description of conditioning would make it easier to understand.

#
The title of the article is "kidney transplantation", but it is not focused on the kidney.

Most of the descriptions are not based on research results obtained from kidneys or kidney-derived cells.

All of the sections in Section 2 should be focused on the kidney.
2. Anesthetic Conditioning
2.1 Prevention of opening of the mPTP 146
2.2 Effect on renal tubular cells and sphingosine-1-phosphate signaling pathway
2.3 Protective effects on the glycocalyx
2.4 Upregulation of hypoxia inducible factors
2.5. Effect on circulating immune cells
2.5.1. Innate immunity
2.5.2 Adaptive immunity
2.5.3. Effect on lymphocyte function antigen-1

#
The phrase "O2" and "CO2" should be "O2" and "CO2".

Author Response

Dear Editor, dear reviewers,

We would like to thank both reviewers for their thorough review of our manuscript. They have addressed some valid points upon which we have adjusted the manuscript accordingly. The adjustments in the manuscript are marked in red. Below you will find a point to point response to the comments of the reviewers. To our opinion the manuscript has improved upon these adjustments and some of the ambiguities are now clarified.

On behalf of the co-authors,

Kind regards,

Gertrude Nieuwenhuijs-Moeke

Reviewer 1

This is a review describing the effects of volatile anesthetic agents on cells.

The authors discuss the effects of VA on cells extensively, although their intention seems to be to focus on the effects on grafts in kidney transplantation.

In this paper, the authors discuss the effects of VA on cells in a wide range of conditions, including pre- (before ischemia), per- (during ischemia) or post- (directly upon reperfusion) conditioning. This is a comprehensive review that includes the latest papers on VA conditioning. But I think there are several issues to be approached by the authors.

The followings are my comments on the manuscript.

#

Explicitly distinguishing between "pre-", "per-", and "post-" in the description of conditioning would make it easier to understand.

Upon a valid request of reviewer 2 we have decided to use the term anesthetic organ protection rather than anesthetic conditioning in several points throughout the manuscript. In paragraph 3 we have added pre- and postconditioning

#

The title of the article is "kidney transplantation", but it is not focused on the kidney.

Most of the descriptions are not based on research results obtained from kidneys or kidney-derived cells.

All of the sections in Section 2 should be focused on the kidney.

  1. Anesthetic Conditioning

2.1 Prevention of opening of the mPTP 146

2.2 Effect on renal tubular cells and sphingosine-1-phosphate signaling pathway

2.3 Protective effects on the glycocalyx

2.4 Upregulation of hypoxia inducible factors

2.5. Effect on circulating immune cells

2.5.1. Innate immunity

2.5.2 Adaptive immunity

2.5.3. Effect on lymphocyte function antigen-1

This is a fair point made by the reviewer. However as reviewer 2 mentioned this is for a good reason as a majority of the basic research has been conducted in the heart, and it would be wrong to omit this important research. We have added several sentences throughout the manuscript to point this out and put this into perspective. We have changed the title of the manuscript and paragraph 3.   

#

The phrase "O2" and "CO2" should be "O2" and "CO2".

We thank the reviewer for noticing this and have replaced 2 by underscore 2

Reviewer 2

The authors are to be congratulated for their attempt to write this comprehensive narrative review about molecular mechanisms of organ protection by volatile anaesthetic.

The review is well structured and overall well written.

The wording in the title and throughout the manuscript of volatile anesthetic conditioning may be misleading. The term conditioning is based on ischaemic pre-conditioning, which has been a historically important mechanism and a milestone within the theme of organ protection. However, throughout the manuscript potential mechanisms of organ protection by volatile agents are described and it would be more appropriate to use the term anesthetic organ protection rather than anesthetic conditioning.

This is a very valid point and we would like to thank the reviewer for pointing this out. We have changed the title to: Molecular Aspects of Volatile Anesthetic Induced Organ Protection and its Potential in Kidney Transplantation. In addition we have replaced AC in organ protection or similar terms in several sentences throughout the  manuscript, including the abstract and title of paragraph 2. The focus now is not on anesthetic conditioning alone but on the overall organ protective properties of VA.    

The review focusses on organ protection by volatile anaesthetics, including the myocardium, the kidneys and the brain. This is for a good reason as a majority of the basic research has been conducted in the heart, and it would be wrong to omit this important research. However, the title appears to imply renal protection in kidney transplantation. Therefore, would a title similar to Molecular Aspects of Anesthetic Organ Protection with a Focus on Kidney Transplantation be more appropriate?

Please see find the adjusted title above. In addition we have added the sentences: Therefore the use of VA during the donation, transplantation and even the preservation period, could be an attractive way to reduce injury and improve graft function. This review highlights the molecular and cellular protective points of engagement of VA shown in in vitro studies and in vivo animal experiments and the potential translation of these results to the clinical setting of kidney transplantation

At the end of the introduction. In addition we have added Animal and in vitro experiments in various organs to the beginning of the summary.

Ischaemia reperfusion injury is a well-described phenomenon, which is not commonly named ischemia and reperfusion injury, this should be amended throughout the manuscript, in order to avoid confusion for the reader.

We have adjusted this in the manuscript

Specific comments

1) Abstract. Page 1 line 19. It should say:… of VA shown in in vitro studies and in vivo animal experiments.

We would like to thank the reviewer for pointing this out, we have corrected this sentence in the abstract and the introduction

2) Page 2 line 49. DAMPS are these damage or danger associated molecular patterns? Please clarify.

Danger associated molecular patterns is the same as damage associated molecular patterns. We have chosen the term danger

3) Page 2 line 50. ...release of DAMPS into intra- and extracellular space. Is there a reference for this statement?

We have added 2 references

Meissner M., Viehmann S. F., Kurts C. DAMPening sterile inflammation of the kidney. Kidney Int. 2019, 95(3), 489-491. https://doi.org/10.1016/j.kint.2018.12.007

Eltzschig H. K., Eckle T. Ischemia and reperfusion--from mechanism to translation. Nat Med. 2011, 17(11), 1391-401. https://doi.org/10.1038/nm.2507

4) Page 2 line 50.  From here several injury cascades are implicated . Why not use the term followed?

If we change from here to followed it would be a too long sentences. Therefore we have remained the original sentences

5) Page 2 line 58. Replace Next to this with In addition.

We have replaced next to this

6) Page 2 line 77. ... called anesthetic conditioning (AC). Replace conditioning with protection. It would be important to include a reference after protection.

We have replaced AC by protective effects of VA. According to the guidelines of the journal we are not allowed to add references in the middle of a sentence.

7) Page 4 Fig 1. This is a busy figure with volatile anaesthetics appearing at the top left hand corner and at the bottom right hand corner. Why? Should effect of VA on Immune cells be included at the top left corner and non-immune related protective effects by VA in the lower right corner?

We agree with the reviewer that this is a busy figure, however, we think it is very informative illustrating the various points of engagement of VA in IRI. We have used an infographic of IRI we’ve made for a review on molecular aspect of IRI and added points of engagement of VA. This figure is an infographic of the manuscript in a nutshell. Adding  more text to it like the reviewer suggested to our opinion makes it less readable.

8) Page 4 line 146. Add Fig 2 behind Prevention of opening of the mPTP

Would the reviewer like us to add fig 2 to the title of a paragraph? We have not seen this before. Already early in the paragraph we  reference to the figure and hope this satisfactory

9) Page 4 line 151. ... activating the innate immune system. A relevant reference should be added here.

We have added to reference

Eltzschig H. K., Eckle T. Ischemia and reperfusion--from mechanism to translation. Nat Med. 2011, 17(11), 1391-401. https://doi.org/10.1038/nm.2507

10) Page 4 line 166. The RISK pathway was first described in detail by Derek J Hausenloy and Derek Yellon in 2004 and this reference should be included.

We have added the reference

11) Page 5 line 171. The SAFE pathway was first described by Sandrine Lecour J Mol Cell Cardiol 2009 and this reference should be included.

We have added the reference

12) Page 6 line 207. The title 2.2. Effect on .... and sphingosine-1-phosphate signalling pathway. (Fig 3) should be included after the title in this line.

We have included and sphingosine-1-phosphate signalling pathway into the title of fig 3

13) Page 6, 2.2. This paragraph would be better placed behind the paragraphs about HIF up-regulation (2.2) and glycocalyx protection (2.3). Furthermore, are any of the mechanisms described in this paragraph included in Fig 2?

We have adjusted the sequence of the paragraphs to: 2.2 glycocalyx protection, 2.3 HIF up-regulation and 2.5 Effect on renal tubular cells. None of the mechanism described in this paragraph (2.5) are included in fig 2.

14) Page 7 line 255. free GAG: it is not clear what is meant by 'free' and 'waves through', please clarify.

We have replaced the original sentences  by the following; Hylaronic acid is the only GAG not covalently attached to a core protein, but instead, interacts with the endothelium by attachment to the transmembrane receptor CD44.[97] It is much longer than the other GAGs and weaves through the glycocalyx.[79]. Waves should have been weaves

15) Page 10 lines 388 and 389. The list of ongoing clinical trials with the NCT references does not provide any information about these potentially interesting trials, particularly if only the NCT registrations are listed. Should these trials be described more in detail and included on page 14 under 3.Clinical Trials?

Since these are mostly trials in oncological surgery these trials to our opinion should not be placed in paragraph 3. We have added some additional information and have removed NCT01367418

Unfortunately these results are not directly translational to the clinical setting but clinical trials looking into the effect of different anesthetic agents or techniques on circulating immune cells, mostly in the field of oncological surgery, are emerging, (NCT03193710, NCT03431532, NCT02567942 ).

16) Page 10 line 389. NCT01367418 is a trial which was completed in 2012, according to the registration website. Should this trial registration be included in this review?

Please see above

17) Page 14 line 552. The heading of Clinical Trials is somehow misleading, particularly as all types of organ protections (heart, brain and kidneys) were discussed in previous chapters, and this chapter is exclusively covering clinical trials in kidney transplantation. Rename the title of chapter 3. to Clinical Trials and Potential Anesthetic Protection in Kidney Transplant Surgery?

We completely agree with this point and have changed the title of the paragraph to the following: Clinical trials on the potential of volatile anesthetic protection in kidney transplantation

Reviewer 2 Report

The authors are to be congratulated for their attempt to write this comprehensive narrative review about molecular mechanisms of organ protection by volatile anaesthetic.

The review is well structured and overall well written.

The wording in the title and throughout the manuscript of volatile anesthetic conditioning may be misleading. The term conditioning is based on ischaemic pre-conditioning, which has been a historically important mechanism and a milestone within the theme of organ protection. However, throughout the manuscript potential mechanisms of organ protection by volatile agents are described and it would be more appropriate to use the term anesthetic organ protection rather than anesthetic conditioning.

The review focusses on organ protection by volatile anaesthetics, including the myocardium, the kidneys and the brain. This is for a good reason as a majority of the basic research has been conducted in the heart, and it would be wrong to omit this important research. However, the title appears to imply renal protection in kidney transplantation. Therefore, would a title similar to Molecular Aspects of Anesthetic Organ Protection with a Focus on Kidney Transplantation be more appropriate?

Ischaemia reperfusion injury is a well-described phenomenon, which is not commonly named ischemia and reperfusion injury, this should be amended throughout the manuscript, in order to avoid confusion for the reader.

Specific comments

1) Abstract. Page 1 line 19. It should say:… of VA shown in in vitro studies and in vivo animal experiments.

2) Page 2 line 49. DAMPS are these damage or danger associated molecular patterns? Please clarify.

3) Page 2 line 50. ...release of DAMPS into intra- and extracellular space. Is there a reference for this statement?

4) Page 2 line 50.  From here several injury cascades are implicated . Why not use the term followed?

5) Page 2 line 58. Replace Next to this with In addition.

6) Page 2 line 77. ... called anesthetic conditioning (AC). Replace conditioning with protection. It would be important to include a reference after protection.

7) Page 4 Fig 1. This is a busy figure with volatile anaesthetics appearing at the top left hand corner and at the bottom right hand corner. Why? Should effect of VA on Immune cells be included at the top left corner and non-immune related protective effects by VA in the lower right corner? 

8) Page 4 line 146. Add Fig 2 behind Prevention of opening of the mPTP

9) Page 4 line 151. ... activating the innate immune system. A relevant reference should be added here.

10) Page 4 line 166. The RISK pathway was first described in detail by Derek J Hausenloy and Derek Yellon in 2004 and this reference should be included.

11) Page 5 line 171. The SAFE pathway was first described by Sandrine Lecour J Mol Cell Cardiol 2009 and this reference should be included.

12) Page 6 line 207. The title 2.2. Effect on .... and sphingosine-1-phosphate signalling pathway. (Fig 3) should be included after the title in this line. 

13) Page 6, 2.2. This paragraph would be better placed behind the paragraphs about HIF up-regulation (2.2) and glycocalyx protection (2.3). Furthermore, are any of the mechanisms described in this paragraph included in Fig 2?

14) Page 7 line 255. free GAG: it is not clear what is meant by 'free' and 'waves through', please clarify.

15) Page 10 lines 388 and 389. The list of ongoing clinical trials with the NCT references does not provide any information about these potentially interesting trials, particularly if only the NCT registrations are listed. Should these trials be described more in detail and included on page 14 under 3.Clinical Trials

16) Page 10 line 389. NCT01367418 is a trial which was completed in 2012, according to the registration website. Should this trial registration be included in this review?

17) Page 14 line 552. The heading of Clinical Trials is somehow misleading, particularly as all types of organ protections (heart, brain and kidneys) were discussed in previous chapters, and this chapter is exclusively covering clinical trials in kidney transplantation. Rename the title of chapter 3. to Clinical Trials and Potential Anesthetic Protection in Kidney Transplant Surgery?

Author Response

(The authors gave the same response as above.)

Round 2

Reviewer 1 Report

I think that the authors did an excellent job of improving the manuscript.

Reviewer 2 Report

All points of my initial review have been addressed satisfactorily and I do not have any more comments.